# Potential Anticancer Effects of Isoflavone Prunetin and Prunetin Glycoside on Apoptosis Mechanisms

**DOI:** 10.3390/ijms252111713

**Published:** 2024-10-31

**Authors:** Se Hyo Jeong, Hun Hwan Kim, Min Yeong Park, Pritam Bhangwan Bhosale, Abuyaseer Abusaliya, Kwang Hyun Hwang, Yeon Gyu Moon, Jeong Doo Heo, Je Kyung Seong, Meejung Ahn, Kwang Il Park, Chung Kil Won, Gon Sup Kim

**Affiliations:** 1Research Institute of Life Science and College of Veterinary Medicine, Gyeongsang National University, Jinju 52828, Republic of Korea; tpgy123@gmail.com (S.H.J.);; 2Biological Resources Research Group, Gyeongnam Department of Environment Toxicology and Chemistry, Korea Institute of Toxicology, 17 Jegok-gil, Jinju 52834, Republic of Korea; 3Korea Institute of Toxicology, 141, Gajeong-ro, Yuseong-gu, Daejeon 35345, Republic of Korea; 4Laboratory of Developmental Biology and Genomics, BK21 PLUS Program for Creative Veterinary Science Research, Research Institute for Veterinary Science, College of Veterinary Medicine, Seoul National University, Seoul 08826, Republic of Korea; 5Department of Animal Science, College of Life Science, Sangji University, Wonju 26339, Republic of Korea

**Keywords:** anticancer, apoptosis, isoflavone, prunetin glucosides, signaling pathway

## Abstract

Cancer is a deadly disease caused by cells that deviate from the normal differentiation and proliferation behaviors and continue to multiply. There is still no definitive cure, and many side effects occur even after treatment. However, apoptosis, one of the programs imprinted on cells, is becoming an important concept in controlling cancer. Flavonoids are polyphenolic compounds found in plants, are naturally bioactive compounds, have been studied for their anticancer effects, and have fewer side effects than chemical treatments. Isoflavones are phytoestrogens belonging to the flavonoid family, and this review discusses in depth the potential anticancer effects of prunetin, one of the many flavonoid families, via the apoptotic mechanism. In addition, a glycoside called prunetin glucoside has been investigated for its anticancer effects through apoptotic mechanisms. The primary intention of this review is to identify the effects of prunetin and its glycoside, prunetin glucoside, on cell death signaling pathways in various cancers to enhance the potential anticancer effects of these natural compounds.

## 1. Introduction

A tumor is a mass of cells that develops as a result of abnormal cell deregulation and soon becomes cancerous and invades surrounding tissue [1]. The causes of cancer are varied and include genetic and environmental factors, as well as habits such as smoking, high fat intake, and UV radiation [2]. Although we still have not completely cured these cancers, many natural products are actually being used in combination with chemotherapy drugs to treat them [3]. In particular, natural bioactive compounds such as flavonoids, which are secondary metabolites of plants, have attracted a lot of attention because they are being actively studied for their anticancer effects and have few side effects [4]. In many studies, flavonoids are believed to target abnormal and damaged cancer cells through the mechanism of apoptosis [4]. This cell death is programmed cell death and is a key mechanism in anticancer therapy [5]. The pro-apoptotic effects of flavonoids are associated with a variety of cell signaling pathways, including mitochondrial function and interactions between cellular components, which in turn regulate the expression of pro-apoptotic proteins [6].

Flavonoids, a class of polyphenolic compounds, are naturally occurring bioactive secondary metabolites found in plants and fungi [7]. They are mainly responsible for color, protecting plants from ultraviolet radiation and acting as signals in the interactions between plants and microorganisms [8]. To date, more than 6000 flavonoids have been identified, and there is still much discussion about their consumption and benefits in humans [9]. Flavonoids are widely used in a variety of food, pharmaceutical, medical, and cosmetic applications and act as powerful antioxidants to protect plants from adverse conditions, and have also been studied for their anti-inflammatory and anticancer properties [4,9]. In addition, many flavonoids are known to exert anticancer effects by modulating cancer development processes such as cell proliferation, apoptosis, and angiogenesis, but the exact signaling mechanisms are not fully understood [10]. Isoflavones, a subgroup of flavonoids, are polyphenolic compounds found in many legumes and have estrogen-like actions due to their structural similarity to the hormone estrogen [11]. Some studies suggest that isoflavones improve cholesterol levels, reduce the risk of osteoporosis, and help with female menopause and heart disease [12]. They also have potential anticancer effects and are being studied in detail [13].

Prunetin, a flavonoid, is an O-methylated isoflavone with a structure based on a 2-phenylchromene-4-skeleton. It is a flavanone derivative and a compound belonging to a subgroup of flavones [14,15]. According to Patel, K. et al., review studies have shown that prunetin has antioxidant and anti-inflammatory properties and potential anticancer effects [16]. However, the specific mechanisms of these effects are not fully understood and require further research. Prunetin glucoside is a glycoside of its precursor prunetin, in which a glucose molecule is attached to the prunetin aglycone form, with the specific binding site determined by the hydroxyl position of the prunetin structure [17]. Recent studies of prunetin glucoside have also identified its potential anticancer effects and are analogous to studies of some flavonoid glycosides that have been found to have antioxidant, anti-inflammatory, and anticancer effects [18].

This review focuses on isoflavones, prunetin, and its glycoside, prunetin glucoside, a subgroup of many flavonoids that have been studied for their cell death mechanisms and effects on cancer. Also, we discuss future research directions for isoflavones, prunetin glycosides, and their potential use in anticancer therapy.

## 2. One Important Strategy as an Antitumor Mechanism Is Apoptosis

### 2.1. Major Tumor Suppression Mechanisms

Cancer is known to be a difficult disease to treat, with many causes and variables that make it difficult to fully cure. There are two main causes of cancer: endogenous factors, such as aging, hormones, growth factors, and inflammation, and exogenous factors, such as radiation, chemical carcinogens, and viruses [19]. Cancer cells are characterized by unlimited proliferation, invasion and metastasis, angiogenesis, and malfunctioning of cell death [20,21].

Apoptosis plays an important role in cancer cell suppression and is a typical example of programmed cell death, along with necrosis and autophagy. This cell death is characterized by a highly regulated program of cell shrinkage, condensation of nuclear chromatin, fragmentation of DNA, and formation of the apoptotic body [22].

### 2.2. Key Proteins Involved in Apoptosis

#### 2.2.1. Cytochrome c

Cytochrome c resides between the mitochondrial membranes and is released into the cytoplasm, nucleus, and extracellular space in response to certain stimuli. Under certain conditions of cellular stress, the mitochondrial outer membrane permeability (MOMP) increases, releasing cytochrome c, which interacts with an apoptosis-inducing factor called APAF1 (apoptotic peptidase activator 1) to form an apoptosome, leading to caspase-mediated cell death [23]. After that, the recruitment and activation of procaspase-9 then initiates the pathway of apoptosis. Recently, Pessoa, J. et al. investigated how cytochrome c can be applied to the development of various anticancer therapies to optimize their apoptotic efficacy and stabilize them for targeted delivery to cancer cells [24].

#### 2.2.2. Bcl-2 Family

The B-cell lymphoma 2 (BCL-2) protein family is also closely related to cell death and includes the anti-apoptotic proteins Bcl-2, Bcl-xL, Bcl-w, and Mcl-1, as well as pro-apoptotic proteins BAX, BAK, and Bim [25]. Primarily, this family regulates mitochondrial outer membrane permeability (MOMP), leading to apoptosis [26]. However, recent studies by King, L.E., et al. have shown that these MOMPs are much more complex than direct regulators and that they are involved in regulatory processes other than apoptosis [27]. For example, activation of BAX and BAK is not necessarily required to bind to BCL-2 and inhibit anti-apoptosis; i.e., activation of BAX and BAK can be induced in the absence of pro-apoptotic BH3 only proteins (Bim, Bid, Bad, Noxa, Puma) that share the BH3 domain, and studies have shown that BAK and BAK alone, without other initiation proteins, can induce MOMPs [27].

#### 2.2.3. P53

P53 is a protein that is encoded and expressed by the TP53 gene and is known to be a cancer suppressor gene. The primary role of P53 is to regulate the cell cycle, allowing normal cell division to occur and promptly inducing cell death when certain stresses are disrupted. If this gene is not working properly due to a mutation, it will divide indefinitely and progress to cancer [28]. Recently, Zhou, S., et al. identified a small molecule compound called H3 that reactivates a mutation that targets Y220C on p53. These compounds induce cell cycle arrest and apoptosis, making them promising candidates for future cancer treatments [29]. Another recent study by Wang, J., et al. focuses on the stabilization of P53 mutations, where molecular chaperones such as Hsp90 and BAG2 stabilize and accumulate P53 mutations and further aggravate tumors. Therefore, researchers have targeted it to destabilize mutations in P53 and induce cancer cell death [30].

#### 2.2.4. Fas and FasL

Fas, also known as CD95 or APO-1, is a cell surface receptor protein that plays an important role in the process of programmed cell death. Fas is a member of the tumor necrosis factor (TNF) receptor superfamily and is expressed on the surface of a variety of cell types, including immune cells, primarily T and B cells [31]. When Fas binds to its Fas ligand (FasL), which is normally expressed on the surface of cytotoxic T cells, it triggers a signal transduction cascade within the cell. This signaling cascade leads to the activation of caspases, a group of protease enzymes whose role is to execute the apoptosis process [32].

#### 2.2.5. Survivin

Survivin is a member of the inhibitor of apoptosis protein (IAP) family. It has a unique expression pattern among IAPs, being highly expressed in embryogenesis and various cancers, whereas it is generally absent or expressed at low levels in most normal adult tissues [33,34]. In apoptosis, survivin may function by inhibiting caspases, enzymes that play a central role in the initiation and execution of cell death. Survivin prevents cells from undergoing apoptosis by inhibiting caspases and instead promotes cell survival [35]. Survivin is primarily known for its anti-apoptotic properties, but there is evidence that it may also have pro-apoptotic properties under certain conditions or in certain cellular contexts. Therefore, Huang, Y.H. and C.T. Yeh said further research is needed to fully elucidate the complex role of survivin in cell death and its potential as a therapeutic target for cancer and other diseases [36].

#### 2.2.6. Caspase Family

Caspases are a family of protease enzymes that play a key role in cell death, also known as programmed cell death. These enzymes are responsible for coordinating the biochemical events that control the breakdown and elimination of cells [37,38]. There are two main groups of caspases involved in cell death [39]. First, initiator caspases are activated in response to apoptotic signals. They initiate the cell death process by cleaving and activating downstream effector caspases. Initiator caspases include caspase-8 and caspase-9 [40]. Effector caspases (also known as executioner caspases) carry out the final steps of cell death once activated by initiator caspases. They cleave specific cellular proteins and substrates, causing characteristic morphological and biochemical changes associated with apoptosis. Effector caspases include caspase-3, caspase-6, and caspase-7 [41]. Another important regulator of cancer cells in angiogenesis, which is a hallmark of many cancers, is VEGF, which also induces cell death through the terminal caspase pathway. Usually, VEGF binds to VEGFR and promotes angiogenesis. A study by Farzaneh Behelgardi, M. et al. showed that VEG4, a representative antagonist, interferes with VEGF/VEGFR interaction, inactivating the PI3K/AKT/XIAP pathway and downregulating the caspase pathway to inhibit apoptosis [42].

### 2.3. Apoptosis Pathway

Apoptosis typically occurs through an intrinsic pathway (also known as the mitochondrial pathway) and an extrinsic pathway (also known as the death receptor pathway) (Figure 1). These pathways can intersect and even mutually regulate each other in a variety of circumstances [43].

#### 2.3.1. Intrinsic Pathway

The intrinsic pathway is initiated by intracellular signals such as DNA damage, oxidative stress, or growth factor mediation [44]. These signals lead to the activation of pro-apoptotic proteins such as Bax and Bak, which promote mitochondrial outer membrane permeabilization (MOMP). They also inactivate the anti-apoptotic proteins Bcl-2 and Bcl-xL. After that, MOMP releases pro-apoptotic proteins from the mitochondria into the cytoplasm, including cytochrome c. Cytochrome c forms an apoptotic complex with other proteins that activates the initiator caspase-9. Activated caspase-9 cleaves and activates downstream effector caspases (e.g., caspase-3, caspase-7) to execute apoptosis [45].

#### 2.3.2. Extrinsic Pathway

The extrinsic pathway is initiated by extracellular signals such as the binding of death ligands to death receptors on the cell surface [46]. There are types of death ligands, including Fas ligand (FasL), tumor necrosis factor alpha (TNF-alpha), and tumor necrosis factor (TNF)-related apoptosis-inducing ligand (TRAIL). The binding of death ligands to receptors (e.g., Fas receptor, TNF receptor, and TRAIL receptor) results in receptor trimerization and recruitment of adaptor proteins such as the Fas-associated death domain (FADD). The adaptor proteins then recruit and activate initiation caspases, such as caspase-8 and caspase-10, to form the death-inducing signaling complex (DISC). Activated initiator caspases in the DISC cleave and activate downstream effector caspases to induce apoptosis [47,48].

## 3. Flavonoids as Bioactive Compounds for Anticancer Effects

Natural bioactive compounds are substances derived from natural resources, such as plants, animals, fungi, and microorganisms, that exhibit biological activity within living organisms. These compounds interact with the physiological systems and can affect a variety of biological processes [49]. These natural bioactive compounds include alkaloids, terpenoids, polyphenols, and flavonoids and are currently under active research for potential therapeutic applications in medicine and healthcare [50,51].

The basic structure of flavonoids, which are double polyphenolic compounds, consists of 15 carbons, two aromatic rings, and one pyran ring, and flavonoids are classified into various subgroups based on changes in the carbon structure of each ring and the degree of oxidation or unsaturation, including flavone, flavonol, isoflavone, anthocyanidin, flavanone, and flavan-3-ol (Figure 2) [52]. These bioactive compounds have potential health-promoting effects through mechanisms of antioxidant, anti-inflammatory, anti-allergic, anticancer, and cardioprotective properties, scavenging free radicals, modulating enzyme activity, inhibiting inflammatory pathways, and modulating cell signaling pathways. However, further research is needed to fully understand their mechanisms of action and therapeutic potential [52].

Various studies have shown that flavonoids arrest the uncontrolled cell cycle, inhibit cancer cell angiogenesis, proliferation, and invasion, and induce autophagy and apoptosis [52,53]. Numerous flavonoids have been shown to lower the incidence of major cancers such as stomach, breast, prostate, and colorectal cancers, and many are candidates for drug selection [53]. It is worth noting that this natural compound, found in a variety of plants, is a researched strategy for treating cancer, and its combination with other compounds is also being explored [54].

## 4. Potential Apoptotic Antitumor Effects of Various Isoflavone Glycosides and Aglycone Forms

Isoflavone glucosides are isoflavones bound to glucose molecules and are a subgroup of isoflavones, a class of phytochemicals found in plants, especially legumes such as soybeans and soy products. Major isoflavone glucosides include genistin, daidzein, and glycitin [55,56].

These glucosides exist in a glycosylated form, which means they are attached to a sugar molecule (glucose) via a glycosidic bond. Upon ingestion of soy products, β-glucosidase hydrolyzes these isoflavone glucosides, dissolving glucose molecules and liberating the non-glycosylated isoflavone forms, daidzein, genistein, and glycitein (Figure 3) [57].

A number of isoflavones have been studied for their potential role in reducing the risk of hormone-related cancers such as breast and prostate cancer, as well as endometrial cancer [58,59].

In the study of Hwang, S.T., et al., treatment with genistin at 50, 100, and 150 μM inhibited estrogen receptor alpha (ERα), activated caspase-8 and 9, and caused cell death through PARP cleavage in MCF-7 cells, a breast cancer cell line, as shown in Table 1 [60].

According to Ravindranath, M.H., et al. and Jung, Y., et al., genistein, the aglycone form of genistin, has been studied for its ability to regulate cell signaling pathways, inhibit cancer cell proliferation, induce apoptosis, and inhibit angiogenesis [61,62]. Multiple studies have shown that genistein inhibits ovarian carcinogenesis and cancer cell growth through multiple mechanisms including ER stimulation, inhibition of cell proliferation, caspase apoptosis, inhibition of angiogenesis, metastasis, and oxidation in SK-OV-3 ovarian cancer cells [63]. In another study by Xu, H., et al., it was also found that 20, 40, and 80 μM treatment of the lung cancer cell line A549 resulted in an increase in caspase-3 and 9 and overexpression of IMPDH2, which restricted the Akt signaling pathway and induced tumor cell apoptosis [64]. In the study by Hsiao, Y.C., et al., on human leukemia HL-60 cancer cells, 20–50 μM treatment induced apoptosis by increasing Bax, PARP cleavage, and caspase-9 and -3, inducing mitochondrial stress, and decreasing anti-apoptotic proteins such as Bcl-2 and Bid (Table 1) [65].

Yao, Z., X. Xu, and Y. Huang discussed how the daidzin treatment of cervical cancer HeLa cells at 20 μM increased mitochondrial membrane permeability and induced apoptosis due to an increase in caspase 8 and 9 (Table 1) [66].

In the study of Kumar, V. and S.S. Chauhan, daidzein, as an aglycone of daidzin, modulated the estrogen signaling pathway and inhibited tumor growth through the overexpression of Bax and inhibition of Bcl2 in MCF-7 breast cancer cells [62,67]. Alshehri, M.M., et al. and Hsu, A., et al. suggested daidzein has also been investigated for its potential role in reducing the risk of breast and prostate cancer by increasing the expression of Bax in MCF-7 and PC3 prostate cancer cells [68,69]. Additionally in the study of Hua, F., et al., daidzein has been shown to have anticancer effects on ovarian cancer through mitochondrial-mediated apoptosis, cell cycle arrest, and inhibition of the Raf/MEK/ERK pathway (Table 1) [70].

As shown in Table 1, Zhang, Y., R. Guo, and W. Yan suggested that glycitin treatment of the lung cancer cell line A549 at 30, 60, and 120 μM resulted in pro-apoptosis with cell membrane damage and nuclear fragmentation [71].

Glycitein, an aglycone of glycitin, increased reactive-oxygen-species-associated apoptosis and mitochondria-associated apoptosis through the MAPK/STAT3/NF-κB pathway and induced G0/G1 cell cycle arrest in AGS gastric cancer cells, as shown by Zang, Y.Q. [62,72]. Lastly, in the study of Zhang, B., et al., glycitein also increased the membrane permeability of human breast cancer SKBR-3 cells, suggesting a possible anticancer effect by damaging the cell membrane [73].

**Table 1 ijms-25-11713-t001:** Apoptosis and antitumor effect of representative isoflavone glycosides and aglycones.

Genistin				
Cancer	Cell Line	Treatment Concentration	Apoptosis Anticancer Mechanism	Reference
**Breast Cancer**	MCF-7	50, 100, and 150 μM	ERα inhibition Caspase-8 and 9 activation PARP cleaved	[60]
**Genistein**				
**Cancer**	**Cell Line**	**Treatment Concentration**	**Apoptosis Anticancer Mechanism**	**Reference**
**Ovarian Cancer**	SK-OV-3	1, 5, 10, 50, and 100 μM	● ER stress ● Caspase-mediated apoptosis ● Inhibition of angiogenesis	[63]
**Lung Cancer**	A549	20, 40, and 80 μM	● Caspase-3 and 9 increased ● Overexpression of IMPDH2 and Akt inhibited	[64]
**Leukemia**	HL-60	20 and 50 μM	● Bax, PARP cleavage, caspase-9 increased ● Mitochondria stress induced ● Bcl-2 and Bid decreased	[65]
**Daidzin**				
**Cancer**	**Cell Line**	**Treatment Concentration**	**Apoptosis Anticancer Mechanism**	**Reference**
**Cervical Cancer**	HeLa	20 μM	● Increases mitochondrial membrane permeability ● Caspase-8 and 9 increased	[66]
**Daidzein**				
**Cancer**	**Cell Line**	**Treatment Concentration**	**Apoptosis Anticancer Mechanism**	**Reference**
**Breast Cancer**	MCF-7	50 µM	● Modulates estrogen signaling pathways ● Overexpression of Bax and inhibition of Bcl2	[67]
**Prostate** **Cancer**	PC3	25 µM	● Bax increased	[69]
**Ovarian** **Cancer**	SK-OV-3	20 µM	● Cytochrome c released ● Caspase-3 and 9 increased ● PARP cleaved ● Raf/MEK/ERK pathway inhibited	[70]
**Glycitin**				
**Cancer**	**Cell Line**	**Treatment Concentration**	**Apoptosis Anticancer Mechanism**	**Reference**
**Lung Cancer**	A549	30, 60, 120 μM	● Cell membrane damage ● Nuclear fragmentation	[71]
**Glycitein**				
**Cancer**	**Cell Line**	**Treatment Concentration**	**Apoptosis Anticancer Mechanism**	**Reference**
**Gastric** **Cancer**	AGS	30 μM	● MAPK/STAT3/NF-κB pathway and mitochondria ● Mediated apoptosis ● Cell cycle arrest	[72]
**Breast Cancer**	SKBR-3	20, 80, and 100 mg/mL	● Membrane permeability increased	[73]

## 5. Potential Apoptotic Antitumor Effects of Prunetin and Prunetin Glycosides

Prunetin is a natural compound that belongs to the flavonoid isoflavone family. This means that, like other isoflavones in particular, it has a chemical structure similar to the hormone estrogen and can exert estrogenic effects in the body [74]. Prunetin has been studied for its potential health benefits and pharmacological activity, and some research suggests that prunetin may have antioxidant, anti-inflammatory, and anticancer properties. In addition, prunetin has been investigated for its potential role in hormone regulation and women’s health, particularly in relation to menopausal symptoms and bone health [16,75].

### 5.1. Apoptosis and Anticancer Effects of Prunetin (In Vitro)

Several studies have shown the anticancer effects of prunetin through apoptosis in a variety of tumors (Table 2). According to Gao, M., et al., treatment of human osteosarcoma MG-63 cells with 20 and 25 μM of prunetin stimulated apoptosis through the enhancement of Bax and caspases, prevented cell proliferation, and decreased Bcl-2 in a dose-dependent manner. It also exhibited antitumor effects by inhibiting peptidyl-prolyl isomerase-1 (Pin-1), which is overexpressed in human malignancies [76]. Köksal Karayildirim, Ç., et al., suggested that prunetin induced apoptosis via caspase3 and TNF-α in bladder cancer RT-4 cells at 21.11 and 42.22 μg/mL [77].

#### Apoptosis and Anticancer Effects of Prunetin (In Vivo)

As shown in Table 3, Yang, X., et al. studied benzo(a)pyrene-induced lung cancer in Swiss albino mice treated orally with prunetin at 30 mg/kg for 12 weeks and 12 to 18 weeks. It decreased carcinoembryonic antigen (CEA) and related metabolites [78]. In another in vivo study by Li, G., et al. in male Wistar rats in which diethylnitrosamine (200 mg/kg) was used to induce liver cancer, treatment with 100 µM/kg of prunetin for 16 weeks delayed liver cancer growth, downregulated cyclin-D1 protein expression, and upregulated Bcl-2, Bax, caspase-3, and caspase-9 gene expression [79].

### 5.2. Apoptosis and Anticancer Effects of Prunetin Glycosides (In Vitro)

#### 5.2.1. Prunetin-4O-Glucoside (Prunetrin)

Prunetin 4′-O-glucoside, also known as 4′-O-beta-D-glucoside and prunetrin, is a glycosyloxyisoflavone. It belongs to the group of 7-methoxyisoflavones and is a hydroxyisoflavone. It shares functional similarities with prunetin. It is mainly found in *Dalbergia sissoo* and *Trifolium pratense* [80] (Figure 4).

Abusaliya, A., et al. suggested that treatment with prunetin 4′-O-glucoside at 10, 20, and 40 μM induced apoptosis in Hep3B liver cancer cells through the activation of PARP and caspase-3 and also affected mitochondrial pathways through increases in caspase-9 and Bak. It also arrested the cell cycle in the G2/M phase and inhibited potential cancer growth by inhibiting the Akt/mTOR pathway [81]. Treatment of HepG2 and Huh7, another liver cancer cell line, with 10, 15, and 30 μM of prunetin 4′-O-glucoside induced intrinsic apoptosis through the release of cytochrome c and activation of caspase-3 and 9, as well as an increase in bak and a decrease in Bcl-xL [82] (Figure 5).

#### 5.2.2. Prunetin-5O-Glucoside (Prunetinoside)

Prunetin 5-O-glucoside, also known as prunetinoside and prunetin 5-O-β-D-glucoside, is a glycosyloxyisoflavone found in *Betula* sp. and *Prunus* sp. [83] (Figure 4). Prunetin 5-O-glucoside has only a few antitumor results, and treatment of AGS gastric cancer cells with 50, 75, and 150 μM of prunetinoside induced apoptosis via PARP and caspase-3. Through network pharmacology Vetrivel, P., et al.’s data suggested that prunetin 5-O-glucoside identified three key protein (CDK2, MMP1, and HSP90) biomarkers that affect gastric cancer, indicating potential anticancer effects [84] (Figure 5).

**Table 2 ijms-25-11713-t002:** Apoptosis and antitumor effects of prunetin and prunetin glycosides (in vitro).

Prunetin				
Cancer	Cell Line	Treatment Concentration	Apoptosis Anticancer Mechanism	Reference
**Osteosarcoma**	MG-63	20 and 25 μM	● Bax and Caspase increased ● Bcl-2 decreased ● Pin-1 inhibited	[76]
**Bladder Cancer**	RT-4	21.11 and 42.22 μg/mL	● Caspase-3 increased ● TNF-α mediated	[77]
**Prunetin 4′-O-glucoside**				
**Cancer**	**Cell Line**	**Treatment Concentration**	**Apoptosis Anticancer Mechanism**	**Reference**
**Liver** **Cancer**	Hep3B	10, 20 and 40 μM	● PARP and caspase-3 activation ● Caspase-9 and Bak increased	[81]
	HepG2 and Huh7	10, 15, 30 μM	● Cytochrome c released ● Bak, Caspase-3 and 9 increased ● Bcl-xL decreased	[82]
**Prunetin 5-O-glucoside**				
**Cancer**	**Cell Line**	**Treatment Concentration**	**Apoptosis Anticancer Mechanism**	**Reference**
**Gastric** **Cancer**	AGS	50, 75 and 150 μM	● PARP and caspase-3 activation	[84]

**Table 3 ijms-25-11713-t003:** Apoptosis and antitumor effects of prunetin (in vivo).

Prunetin			
Cancer	Mouse Model and Dosages	Apoptosis Anticancer Mechanism	Reference
**Lung** **Cancer**	Swiss albino mouse, Benzo(a)pyrene, gavage, 30 mg/kg 12 and 12~18 weeks	● Carcinoembryonic antigen (CEA) decreased	[78]
**Liver** **Cancer**	Male Wistar rat, diethylnitrosamine, feed with water, 200 mg/kg	● Cyclin-D1 downregulated ●Bcl-2, Bax, caspase-3, and caspase-9 upregulated	[79]

## 6. Discussion and Conclusions

This review details how prunetin and prunetin glycosides, like other isoflavones, have been investigated for their potential anticancer effects in various preclinical studies. While research into the specific anticancer properties of prunetin is ongoing and not as extensive as that on other isoflavones such as genistein and daidzein, some studies suggest that these isoflavone glycosides may exert their anticancer effects through multiple mechanisms [81,82,84]. One notable difference compared to other flavonoids is that isoflavone families, such as prunetin or prunetin glycoside, are heavily studied in relatively hormone-sensitive tumors such as breast, ovarian, and prostate cancers [58,59,85,86]. The reason for this is that, as mentioned earlier, the isoflavone family has structural similarities to the hormone estrogen [11]. Another suggestion is whether there is a difference between the aglycone form and the glycoside in anticancer. Glycosylation usually results in higher solubility than typical flavonoids due to increased structural stability, which in turn can lead to greater absorption [87,88]. This characteristic might be one advantage of the glycoside form over the aglycone form in exerting anticancer effects. However, no comparative studies have been conducted between aglycones and glycosides, so more research is needed. Similarly, research on the anticancer effects of prunetin glycosides is promising, but further studies, including more in vitro studies and clinical trials, are needed to confirm definitive efficacy and safety.

## Figures and Tables

**Figure 1 ijms-25-11713-f001:**
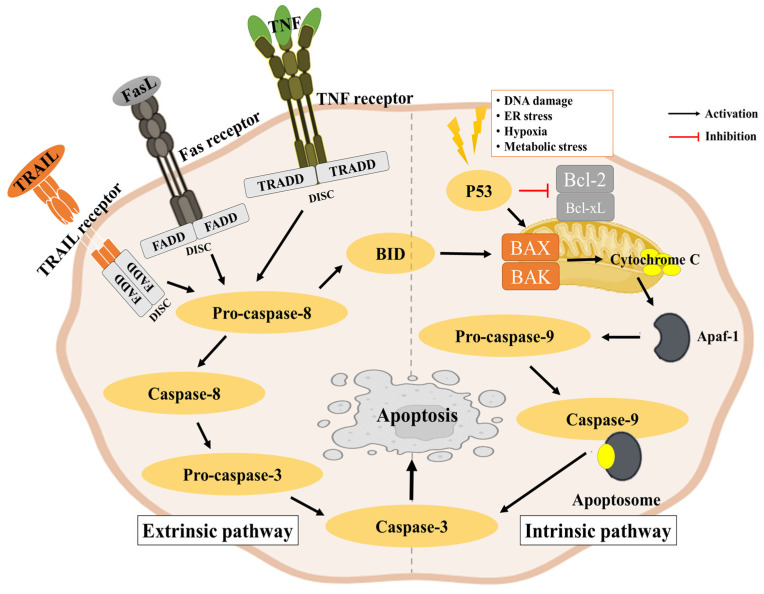
Intrinsic and extrinsic pathways of apoptosis.

**Figure 2 ijms-25-11713-f002:**
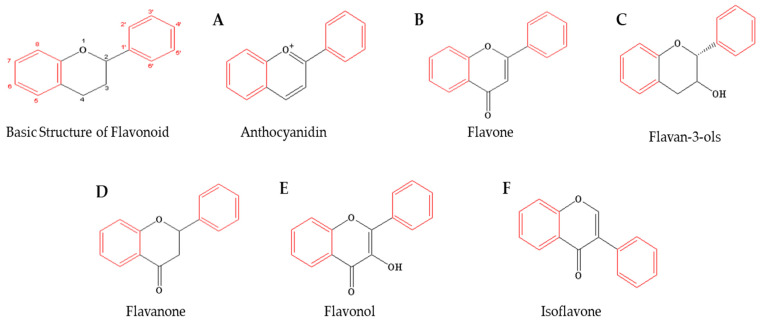
Basic structures of flavonoids and their sub-classes.

**Figure 3 ijms-25-11713-f003:**
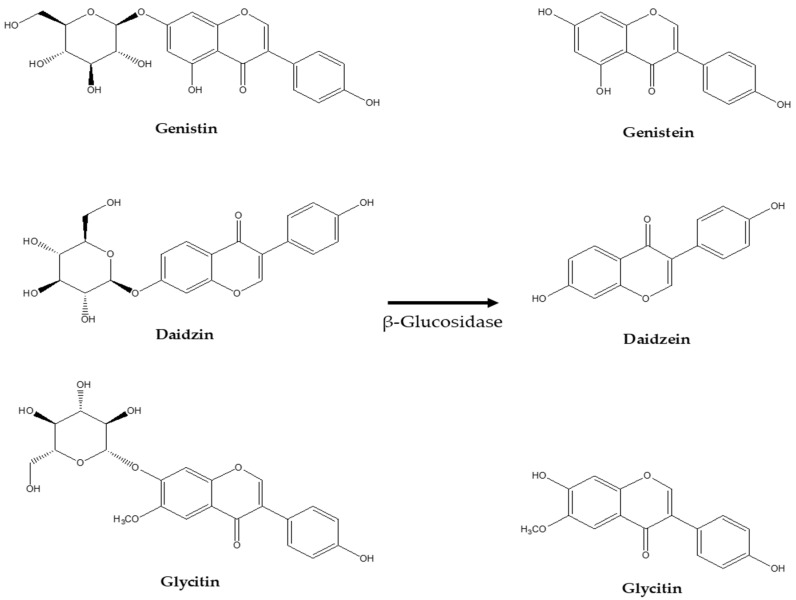
Glycoside and aglycone forms of various isoflavones.

**Figure 4 ijms-25-11713-f004:**
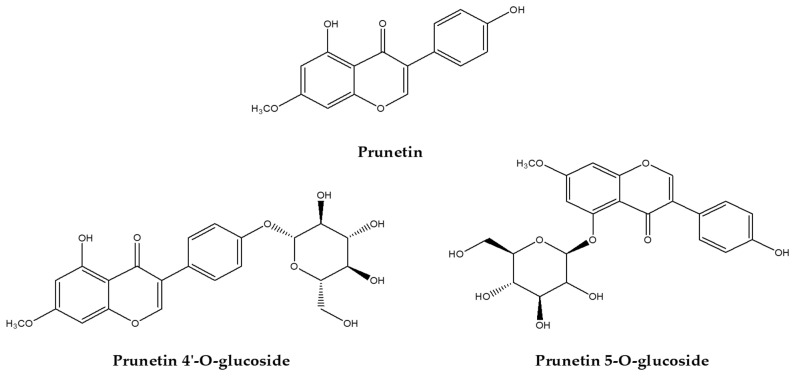
Prunetin and prunetin glycoside structures.

**Figure 5 ijms-25-11713-f005:**
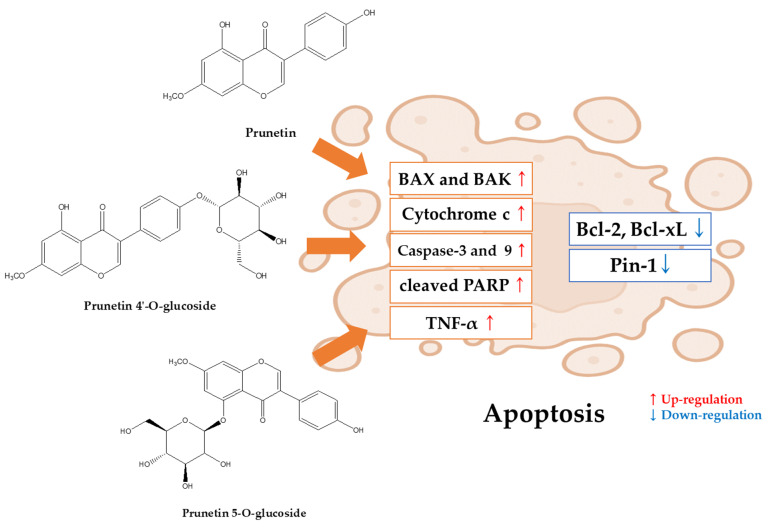
Apoptosis mechanisms of prunetin and prunetin glycosides.

## Data Availability

Samples are not available from the authors.

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
