# Peer review of "Potential Anticancer Effects of Isoflavone Prunetin and Prunetin Glycoside on Apoptosis Mechanisms"

_ijms, 2024, doi:10.3390/ijms252111713_

Round 1
Reviewer 1 Report
Comments and Suggestions for Authors
In this manuscript, Jeong et al. have reviewed the anti-tumor effects of isoflavone prunetin and prunetin glycoside and their apoptosis mechanisms. Their work is helpful in advancing our understanding of the role of isoflavone prunetin and prunetin glycoside in cancer treatments with apoptotic mechanisms. However, before the manuscript is accepted for publishing, the reviewer suggests the authors revise the manuscript as follows:
1. The title of section 2 (Anti-tumor and Apoptosis Mechanism) is too broad and should be revised. This section just briefly introduces apoptosis and its mechanisms in cancer treatments.
2. Since key proteins, such as cytochrome C, Bcl-2 family, P53, Fas, et al., are significant in apoptosis, the reviewer suggests the authors add all these proteins in Figure 1. Meanwhile, their roles (e.g., upregulation or downregulation; activation or inhibition) in apoptosis should be explicit.
3. TNF, FasL, and Trail are three critical extrinsic pathways correlated with apoptosis that have been widely discussed. Since the authors have shown the TRAIL pathway in Figure 1, some discussion should be added in subsection 2.3.2. (Extrinsic Pathway).
4. There is a mistake in Figure 1. Is the interaction between the FasL and TRAIL receptor the primary mechanism that initiates apoptosis through the TRAIL receptor?
5. In section 3 (Flavonoid as Bioactive Compounds), the authors should stress the roles of Falvonoid in cancer treatments, which will fit the aim of this manuscript.
6. What are the differences and similarities between prunetin and prunetin glycosides in inducing apoptosis? What are the differences and similarities between these compounds and other Flavonoids with anti-cancer capacities in inducing apoptosis? These comparisons should be added to the manuscript to highlight the importance of this review.
7. The image resolutions in Figures 2 and 3 are too low. It is recommended that all fonts in the Figures be in the same style.
8. The manuscript should be carefully checked to remove typos and other mistakes.
Author Response
For reviewer 1
Thank you for your attentive comments.
- The title of section 2 (Anti-tumor and Apoptosis Mechanism) is too broad and should be revised. This section just briefly introduces apoptosis and its mechanisms in cancer treatments.
Answer: Thank you for the good point. In section 2 (Anti-tumor and Apoptosis Mechanism), we decided to describe the overall anti-tumor and apoptosis process in an integrated way, but it seemed to be too general, so we narrowed the title a bit more as per the reviewer's suggestion.
Section2 title changed.
“Anti-tumor and Apoptosis Mechanism” à “One Important Strategy as an Anti-tumor Mechanism is Apoptosis”
- Since key proteins, such as cytochrome C, Bcl-2 family, P53, Fas, et al., are significant in apoptosis, the reviewer suggests the authors add all these proteins in Figure 1. Meanwhile, their roles (e.g., upregulation or downregulation; activation or inhibition) in apoptosis should be explicit.
Answer: Thank you for the reviewer's comments. We have added other important proteins to Figure 1, such as cytochrome C, Bcl-2 family, P53, and Fas that you listed. We have also shown the activation or inhibition of proteins in the apoptosis process in Figure 1.
- TNF, FasL, and Trail are three critical extrinsic pathways correlated with apoptosis that have been widely discussed. Since the authors have shown the TRAIL pathway in Figure 1, some discussion should be added in subsection 2.3.2. (Extrinsic Pathway).
Answer: Thanks for the good point . We've added it to Subsection 2.3.2 for theTRAIL pathway .
“There are types of death ligands, including Fas ligand (FasL), tumor necrosis factor alpha (TNF-alpha) and tumor necrosis factor (TNF)-related apoptosis-inducing ligand (TRAIL). Binding of death ligands to receptors (e.g., Fas receptor, TNF receptor, and TRAIL receptor) results in receptor trimerization and recruitment of adaptor proteins such as the Fas-associated death domain (FADD).”
- There is a mistake in Figure 1. Is the interaction between the FasL and TRAIL receptor the primary mechanism that initiates apoptosis through the TRAIL receptor?
Answer: As pointed out by the reviewer, there was an error in FasL, so we have corrected the ligand that binds to the TRAIL receptor in Figure 1 to TRAIL.
- In section 3 (Flavonoid as Bioactive Compounds), the authors should stress the roles of Flavonoid in cancer treatments, which will fit the aim of this manuscript.
Answer: We appreciate the reviewer's advice. In section 3 (Flavonoids as Bioactive Compounds), we have added about the role of flavonoids in cancer treatment.
“Various studies have shown that flavonoids arrest the uncontrolled cell cycle, inhibit cancer cell angiogenesis, proliferation and invasion, and induce autophagy and apoptosis. Numerous flavonoids have been shown to lower the incidence of major cancers such as stomach, breast, prostate, and colorectal cancers, and many are candidates for drug selection. It's worth noting that this natural compound, found in a variety of plants, is a researched strategy for treating cancer, and its combination with other compounds is also being explored.”
- What are the differences and similarities between prunetin and prunetin glycosides in inducing apoptosis? What are the differences and similarities between these compounds and other Flavonoids with anti-cancer capacities in inducing apoptosis? These comparisons should be added to the manuscript to highlight the importance of this review.
Answer: We appreciate the reviewer's point. We agree with the reviewer that identifying the differences in cell death between the aglycone and glycoside forms and how they differ from other flavonoids will be the focus of this manuscript. Although there are not many studies on prunetin and glycosides and none have shown differences between the two parts, we have added relevant studies and the authors' comments to the manuscript and revised the title of Section 6 to Discussion and Conclusions
“One notable difference compared to other flavonoids is that isoflavone families, such as prunetin or prunetin glycoside, are heavily studied in relatively hormone-sensitive tumors such as breast, ovarian, and prostate cancers. The reason for this is that, as mentioned earlier, the isoflavone family has structural similarities to the hormone estrogen. Another suggestion is whether there is a difference between the aglycone form and the glycoside in anticancer. Glycosylation usually results in higher solubility than typical flavonoids due to increased structural stability, which in turn can lead to greater absorption. This characteristic might be one advantage of the glycoside form over the aglycone form in exerting anti-cancer effects. However, no comparative studies have been conducted between aglycones and glycosides, so more research is needed.”
- The image resolutions in Figures 2 and 3 are too low. It is recommended that all fonts in the Figures be in the same style.
Answer: We updated image resolutions in Figures 2 and 3 and all fonts in the Figures changed in the same style.
- The manuscript should be carefully checked to remove typos and other mistakes.
Answer: All typos mistakes have been identified and fixed.

Reviewer 2 Report
Comments and Suggestions for Authors
The paper entitled "Potential Anticancer Effect of Isoflavone Prunetin and Prunetin Glycoside on Apoptosis Mechanism" is an extensive and informative review about the potential antitumor effect of prunetin and prunetin glycoside. The need for new antitumor natural molecules is constantly increasing making research in the field, valuable. I have to suggest minor changes in the text.
1. Line 65, 77 please correct
2. Fig 2 is of low quality, please resize and replace.
3. The authors should state the methodological approach about litterature searching writting this review. Do they include all the recent bibliography? From when and which are the sources? Do they also include all the in vivo experiments done on cancer cells?
4. Since the authors discuss the possible Apoptosis and Anti-Cancer Effects of Prunetin and glycosides, they should also refer to VEGF pathway, which is also implicated.
Author Response
For reviewer 2
Thank you for your attentive comments.
- Line 65, 77 please correct
Answer: Thank you. I corrected the following two lines.
Line 65 “Also, we will also discuss future research directions for isoflavones, prunetin glycosides, and their potential use in anticancer therapy.”
Line 77 “This cell death is characterized by a highly regulated program of cell shrinkage, condensation of nuclear chromatin, fragmentation of DNA, and formation of the apoptotic body”
- Fig 2 is of low quality, please resize and replace.
Answer: Thank you. I've adjusted the quality of Fig2 and replaced it.
- The authors should state the methodological approach about litterature searching writting this review. Do they include all the recent bibliography? From when and which are the sources? Do they also include all the in vivo experiments done on cancer cells?
Answer: For facts that have become so well understood over time (the concept of apoptosis, the concept of flavonoids, etc.), we did not write a special author or approach to literature information.
However, where there is a particular emphasis, where we think it's important, or where facts or information have recently come to light, we've included clear authorship and sources in each sentence, rather than vague sources.
We have also rewritten the experiments on cancer cells to clearly distinguish between in vitro and in vivo experiments. Thank you very much for the good points.
- Since the authors discuss the possible Apoptosis and Anti-Cancer Effects of Prunetin and glycosides, they should also refer to VEGF pathway, which is also implicated.
Answer: Thank you for this great additional comment. We have also added a correlation between Vascular Endothelial Growth Factor, (VEGF), and APOPTOSIS in 2.2.6.
“Another important regulator of cancer cells in angiogenesis, which is a hallmark of many cancers, is VEGF, which also induces cell death through the terminal caspase pathway. Usually, VEGF binds to VEGFR and promotes angiogenesis. A study by Farzaneh Behelgardi, M. et al. showed that VEG4, a representative antagonist, interferes with VEGF/VEGFR interaction, inactivating the PI3K/AKT/XIAP pathway and downregulating the caspase pathway to inhibit apoptosis”

Round 2
Reviewer 1 Report
Comments and Suggestions for Authors
All my concerns have been addressed . I recommend the acceptance of the publishing of this work as it is.